# Mechanical stimulation promotes enthesis injury repair by mobilizing *Prrx1*+ cells via ciliary TGF-β signaling

Han Xiao[1,2,3,4,5], Tao Zhang[1,2,3,4,6], Changjun Li[6,7], Yong Cao[2,3,4,6,8], Linfeng Wang[1,2,3,4,6], Huabin Chen[1,2,3,4,6], Shengcan Li[1,2,3,4,6], Changbiao Guan[1,2,3,4,6], Jianzhong Hu[2,3,4,6,8], Di Chen[9], Can Chen[2,3,4,6,10]*, Hongbin Lu[1,2,3,4,6]*

[1]Department of Sports Medicine, Xiangya Hospital, Central South University, Changsha, China; [2]Key Laboratory of Organ Injury, Aging and Regenerative Medicine of Hunan Province, Changsha, China; [3]Xiangya Hospital-International Chinese Musculoskeletal Research Society Sports Medicine Research Centre, Changsha, China; [4]Hunan Engineering Research Center of Sport and Health, Changsha, China; [5]Department of pediatric orthopedic, Hunan Children's hospital, Changsha, China; [6]National Clinical Research Center for Geriatric Disorders, Xiangya Hospital, Central South University, Changsha, China; [7]Department of Endocrinology, Xiangya Hospital, Central South University, Changsha, China; [8]Department of Spine Surgery, Xiangya Hospital, Central South University, Changsha, China; [9]Faculty of Pharmaceutical Sciences, Shenzhen Institute of Advanced Technology, Chinese Academy of Sciences, Shenzhen, China; [10]Department of Orthopedics, Xiangya Hospital, Central South University, Changsha, China

*For correspondence:
chencanwow@foxmail.com (CC);
hongbinlu@hotmail.com (HL)

**Competing interest:** The authors declare that no competing interests exist.

**Abstract** Proper mechanical stimulation can improve rotator cuff enthesis injury repair. However, the underlying mechanism of mechanical stimulation promoting injury repair is still unknown. In this study, we found that *Prrx1*+ cell was essential for murine rotator cuff enthesis development identified by single-cell RNA sequence and involved in the injury repair. Proper mechanical stimulation could promote the migration of *Prrx1*+ cells to enhance enthesis injury repair. Meantime, TGF-β signaling and primary cilia played an essential role in mediating mechanical stimulation signaling transmission. Proper mechanical stimulation enhanced the release of active TGF-β1 to promote migration of *Prrx1*+ cells. Inhibition of TGF-β signaling eliminated the stimulatory effect of mechanical stimulation on *Prrx1*+ cell migration and enthesis injury repair. In addition, knockdown of *Pallidin* to inhibit TGF-βR2 translocation to the primary cilia or deletion of *Ift88* in *Prrx1*+ cells also restrained the mechanics-induced *Prrx1*+ cells migration. These findings suggested that mechanical stimulation could increase the release of active TGF-β1 and enhance the mobilization of *Prrx1*+ cells to promote enthesis injury repair via ciliary TGF-β signaling.

## Editor's evaluation

The murine enthesis injury model was used investigate the mechanism of proper mechanical stimulation for enthesis injury repair. Mechanical stimulation could increase the release of active TGF-β1 and enhance mobilization of Prx1+ cells to promote enthesis injury repair via ciliary TGF-β signaling. This work is very significant and will provide an excellent advance in the field.

## Introduction

Rotator cuff (RC) tear is common in modern sports activity, which often causes persistent shoulder pain and dysfunction (*Meislin et al., 2005*). Surgical repair has been a well-established and commonly accepted treatment for severe RC tear, especially when conservative treatment fails (*Galatz et al., 2001*; *Chung et al., 2013*). It has been reported that approximately 450,000 patients were accepted RC repairs surgery annually in the United States (*Thigpen et al., 2016*). However, the results of surgical repair are not always satisfactory (*Lee et al., 2020*). Previous studies have shown structural failure of the RC repair ranging from 16% to 94%, with poor outcomes associated with failed microstructure regeneration of RC enthesis (*Thigpen et al., 2016*; *Hein et al., 2015*; *Galatz et al., 2004*). Histologically, the RC enthesis has been consisted of four distinct yet continuous tissue layers: bone, calcified cartilage, uncalcified cartilage, and tendon. This kind of structure can subtly transfer force from muscle to bone, while enthesis functional injury repair remains an insurmountable challenge in sports medicine.*Chen et al., 2019* Therefore, how to promote regeneration of the enthesis is an urgent problem for clinicians.

Moderate mechanical load is essential for enthesis development and maintenance (*Thampatty and Wang, 2018*). Meanwhile, the clinical application of mechanobiological principles following enthesis injury forms the basic rehabilitation protocols (*Thomopoulos et al., 2007*; *Galloway et al., 2013*). However, there is a debate about the initiating time and strength of mechanical stimulation during enthesis healing procedure (*Chang et al., 2015*). In clinical treatment, a traditional rehabilitation program after RC repair has been suggested to delay mechanical exercise (immobilization for about 6 weeks) (*Keener et al., 2014*), while an accelerated protocol suggests that an immediate exercise with limited range of motion would be better for tendon healing (*Lee et al., 2012*). Zhang et al adopted treadmill training at postoperative day 7 on murine enthesis injury repair model and found that mechanical stimulation could improve enthesis fibrocartilage and bone regeneration and obtained better mechanical parameters (*Zhang et al., 2021*). Although we know that there is a correlation between appropriate mechanical stimulation and high quality of enthesis healing, the uncovering mechanism is poorly understood. Revealing the cellular and molecular processes of enthesis healing with mechanical stimulation after surgical repair will allow clinicians to implement preventative interventions and prescribe proper therapeutics to improve clinical outcomes.

During the wound regeneration procedure, soluble inflammatory mediators bind to cell surface or cytoplasmic receptors, and lead to recruitment of immune cells, stem cells and tissue-resident cells by activating signaling cascades (*Wynn and Vannella, 2016*). As we known, stem cells are essential for enthesis regeneration. *Prrx1* is a paired-related homeobox gene that is expressed in undifferentiated mesenchymal stem cells in the developing limb buds (*Kawanami et al., 2009*). A previous study showed that *Prrx1* transgene marked osteochondral progenitors in the periosteum and played an essential role in skeletal development (*Ouyang et al., 2014*). Mice lacking *Prrx1* transgenes would show craniofacial defect, limb shortening, and incompletely penetrant Spina bifida (*Martin and Olson, 2000*). Considering *Prrx1*$^+$ cells are indispensable stem cell lineage for the musculoskeletal system, we want to specificlly reveal the role of *Prrx1*$^+$ cells in enthesis injury repair and uncover the mechanism of mechanical stimulation on improving enthesis injury repair in this study.

Primary cilia is an antenna-like sensory organelle based on immotile microtube, and present on nearly every cell type, including mesenchymal stem cells, endothelial cells (ECs), epithelial cells, fibroblasts, and other cells in vertebrates (*Nauli et al., 2008*; *Bowie and Goetz, 2020*; *Hilgendorf et al., 2019*; *Vion et al., 2018*). Primary cilia contains a distinct subset of receptors and other proteins, which make it a sophisticated signaling center functioning as mechanosensor and chemosensation (*Nauli et al., 2003*; *Hua and Ferland, 2018*; *Guemez-Gamboa et al., 2014*; *Anvarian et al., 2019*). Previous study also found that translocating receptors to the primary cilia could enhance the signaling transmission (*Zheng et al., 2018*). Defect or dysfunction of primary cilia could lead to severe disorders of the body, which is known as ciliopathies, such as polycystic kidney disease, primary ciliary dyskinesia, retinopathies, combined developmental deficiencies, and other sensory disorders (*Bisgrove and Yost, 2006*; *Miyamoto et al., 2020*; *Bergmann et al., 2018*; *May-Simera et al., 2018*; *Robichaux et al., 2019*). Genetic deletion of *Ift88*, an encoded protein closely associated with cilia formation and maintenance, could decrease the load-induced bone formation (*Moore et al., 2018*). At the same time, the primary cilia is a hub for transducing biophysical and hedgehog signals to regulate tendon enthesis formation and adaptation to loading (*Yuan and Yang, 2015*; *Fang et al., 2020*). Therefore,

we wonder if primary cilia also plays an important role in mechanical stimulation signal transmission during enthesis healing procedure.

In this study, we first revealed the characteristics of *Prrx1*+ cells in the developing enthesis by single-cell RNA sequencing (scRNA-seq) and examined the dynamic pattern of *Prrx1*+ cells in murine RC enthesis at different ages. Then, we used the murine enthesis injury model to find out the mechanism of proper mechanical stimulation on stimulating *Prrx1*+ cell migration and enthesis injury repair. Our data demonstrated that appropriate mechanical stimulation could increase the release of active TGF-β1 and enhance mobilization of *Prrx1*+ cells to promote enthesis injury repair via ciliary TGF-β signaling.

## Results

### scRNA-seq analysis reveals the cell populations in the developing enthesis

To determine the cellular composition of the developing enthesis, we isolated and sequenced transcriptomes of individual live CD45-Ter119- cells from the murine enthesis (from the fully patterned limb at E15.5 to early maturity of enthesis with appearance of a modest 4-zone structure at P28) based on 10 × Genomics system (*Figure 1a and b*). After sequencing and data quality processing, we got high-quality transcriptomic data from 21,532 single-cells, including 8,919 E15.5 cells, 7489 P7 cells, and 5124 P28 cells. The single-cell RNA sequencing (scRNA-seq) data had high read depth for most of the single cell samples (*Figure 1—figure supplement 1*). We carried out unbiased clustering analysis for all single-cells and identified 23 major cell populations in the developing enthesis by Seurat analysis (*Figure 1c*, *Figure 1—figure supplement 1*). Through differential gene expression analysis, we annotated 15 clusters into distinct cell types or states based on the expression of genes uniquely or in combinations represented individual cluster identities (*Figure 1—figure supplement 2*). Feature plot showed the canonical marker genes, which enriched in seven enthesis related clusters: BMSCs, Fibroblastic cells, Tenocytes, Chondrocytes, Proliferative stromal cells, Osteoblast/Osteocyte (OB/Ocy), Lepr+ cells (*Figure 1d*). Cell fraction related to enthesis development showed that the rate of OB/Ocy and Lepr+ cells increased significantly in P7 and P28, which was consistent with the enthesis mineralization procedure. At the same time, chondrocytes were high in E15.5, P7 and decreased significantly in P28 (*Figure 1e*). These data suggested that the maturation of enthesis was higly correlated with osteochondrogenesis procedure.

### scRNA-Seq distinguishes *Prrx1*+ cells during enthesis development

To examing the role of of *Prrx1*+ cells in enthesis development, 5,607 *Prrx1*+ cells from mouse enthesis (E15.5, P7, and P28) were analyzed and were grouped into five distinct clusters: BMSCs, Fibroblastic cells, Tenocytes, Chondrocytes, and Lepr+ cells. *Prrx1* expression was relatively high in E15.5 and P7, while decreased significantly in P28 (*Figure 2a*). Pseudotime ordering of *Prrx1*+ cells from the enthesis related five clusters were reconstructed by Monocle, an unsupervised algorithm (*Figure 2b*). The trend of reconstructed trajectory was consistent with the time point (*Figure 2c* upper panel), which could represent the temporal (stem/progenitor and teno/oseto-chondro lineage) relationships during the development of enthesis. The reconstructed trajectory tree colored by clusters shows some overlap along the pseudotime (*Figure 2c* lower panel). These results indicated that *Prrx1*+ cells were highly involved in enthesis development via differentiating into tenocytes, osteoblasts/osteocytes or chondrocytes. To further analyze differential gene expression of *Prrx1*+ cells in E15.5, P7, and P28, GO enrichment analysis was performed and representative GO terms in represented biological processes were illustrated (*Figure 2d*). The results showed that the ribonucleoprotein complex biogenesis and assembly activities were significantly upregulated in E15.5, while extracellular matrix organization activities in P7 and ossification activities in P28. These data suggested that *Prrx1*+ cells could be the potential reliable progenitors for enthesis development.

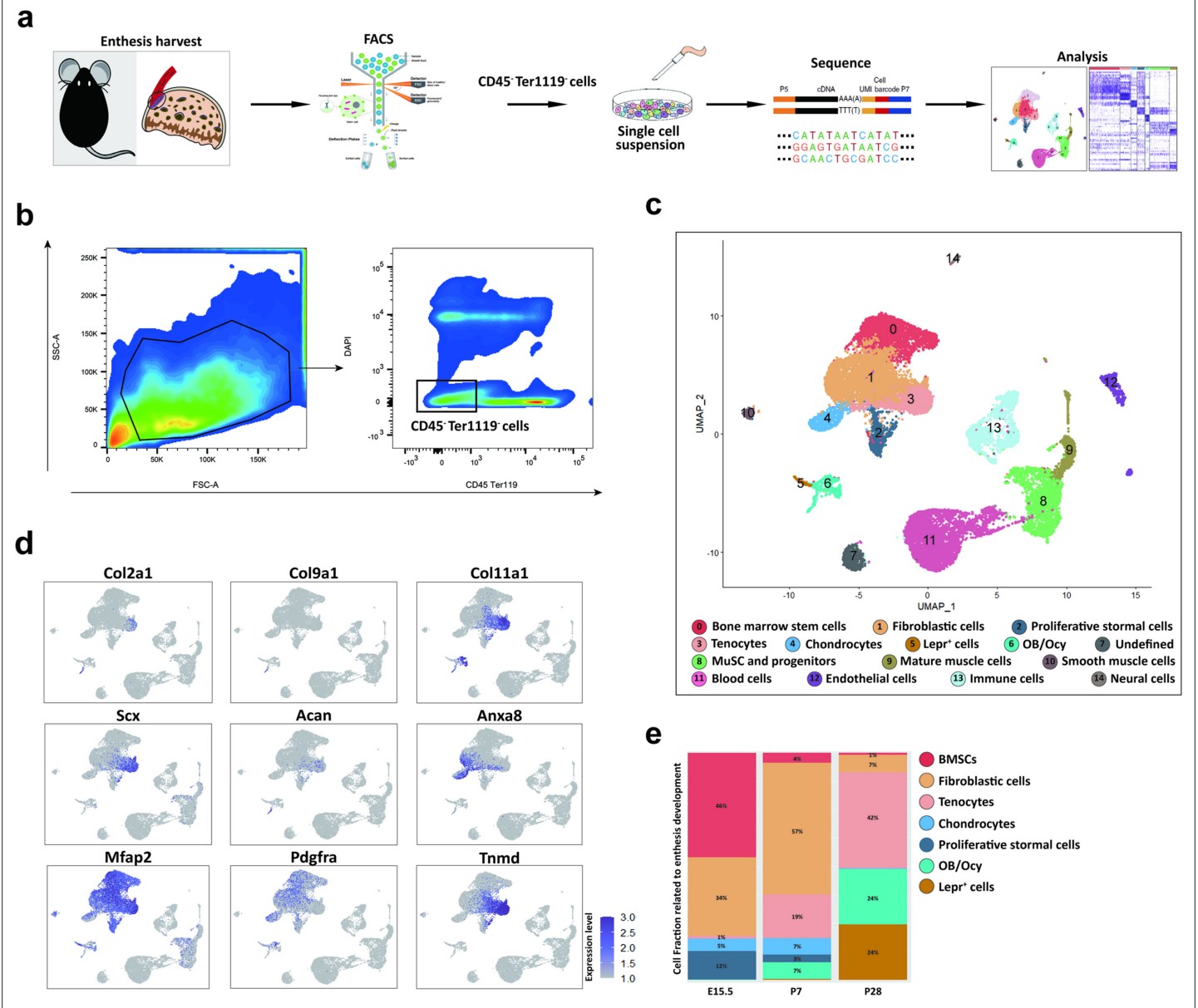

**Figure 1.** scRNA-seq analysis reveals the cell populations in the developing enthesis. (**a**) The flow chart of scRNA-seq analysis. (**b**) Isolation of CD45⁻
Ter119⁻ cells by FACS. (**c**) All cell clusters visualized with uniform manifold projection (UMAP). (**d**) Feature plot of canonical marker genes enriched in
clusters defines enthesis related clusters. (**e**) Enthesis development related cell composition at E15.5, P7, and P28.

The online version of this article includes the following source data and figure supplement(s) for figure 1:

**Source data 1.** The single-cell matrix data of E15.5 mouse TBI.

**Source data 2.** The single-cell matrix data of 1 w mouse TBI.

**Source data 3.** The single-cell matrix data of 4 w mouse TBI part I.

**Source data 4.** The single-cell matrix data of 4 w mouse TBI part II.

**Figure supplement 1.** Quality control of unbiased scRNA-seq dataset.

**Figure supplement 2.** Gene expression defining clusters of E15.5, P7, P28 canonical correlation analysis.

## Lineage tracing of *Prrx1⁺* cells in murine rotator cuff enthesis development and injury repair

To understand the dynamic pattern of *Prrx1⁺* cells in rotator cuff enthesis, we performed immunos-
taining of the murine humeral head, using *Prrx1^CreER-GFP* transgenic mice at P7, P28, and P56. We found

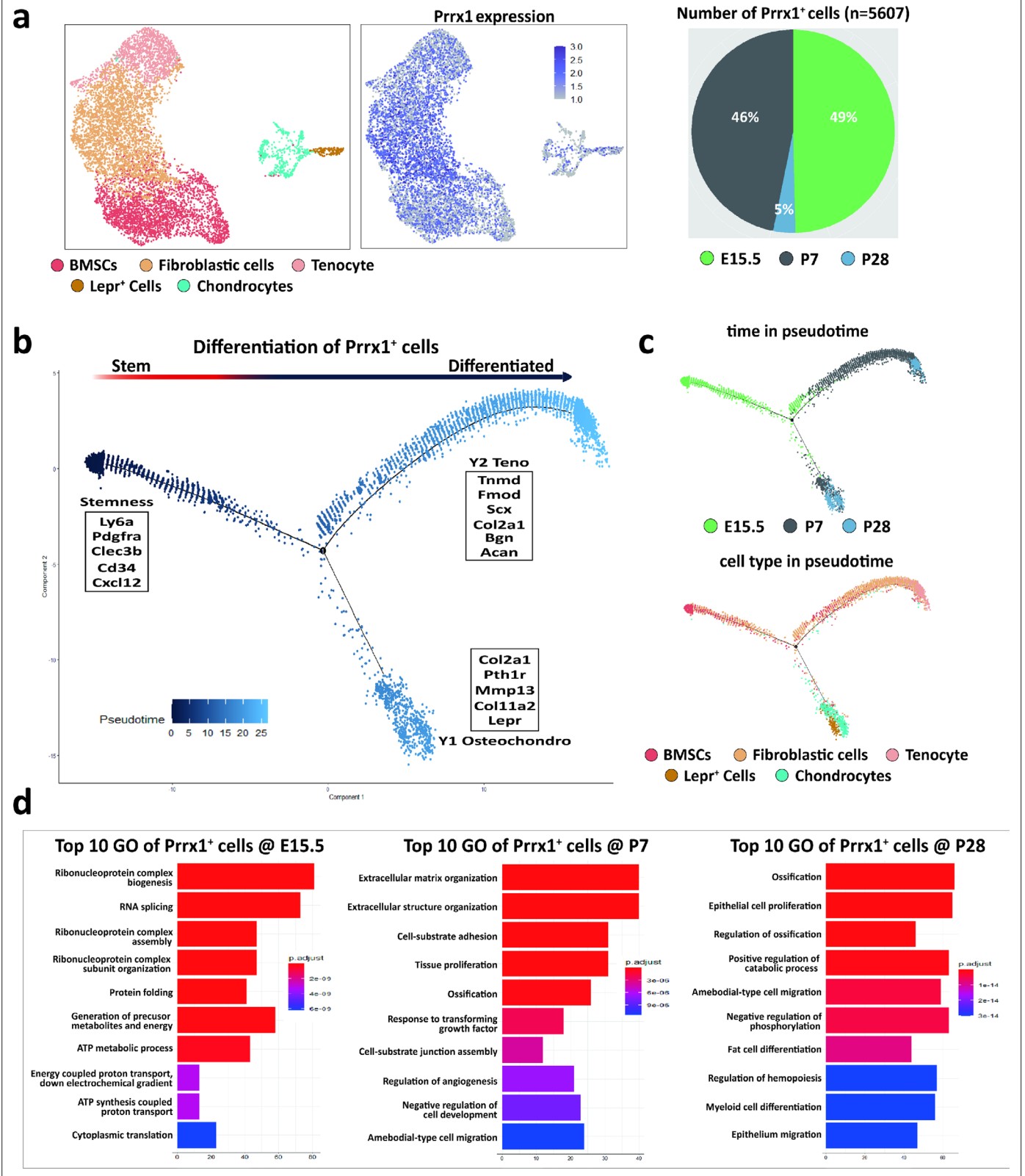

**Figure 2.** scRNA-Seq distinguishes *Prrx1*+ cells during enthesis development. (**a**) 5607 *Prrx1*+ cells from mouse supraspinatus enthesis (E15.5, P7, and P28) were grouped into five distinct clusters (colors indicated). Each point represents an individual cell; Right panel shows the expression level of *Prrx1* within enthesis-related clusters. (**b**) Differentiation trajectory of enthesis-related cells constructed by Monocle and was colored by pseudotime order. Branches on the 2D trajectory tree are indicated as tenogenic branch (**Y1**) and osteochondrogenic branch (**Y2**). (**c**) Upper panel was colored by real

*Figure 2 continued on next page*

Figure 2 continued

time-point and lower panel was colored by cell clusters, respectively. (**d**) The enriched GO terms (biological processes) of differentially expressed genes in enthesis development related *Prrx1*+ + at E15.5, P7, and P28, respectively.

that active *Prrx1*+ cells (high expression of GFP) were abundant on the peripheral humeral head in young mice and decreased markedly during late adulthood (***Figure 3a***). At the enthesis, we found active *Prrx1*+ cells were present during the early postnatal period, while they decreased significantly with age, then, disappeared and were confined within the perichondrium at adulthood (***Figure 3b and c***). To investigate the degree of *Prrx1*+ cells participating in enthesis development at a different age, we generated *Prrx1^CreER*; *Rosa26^tdTomato* mice to permanently label the cells coming from *Prrx1*+ cell pool. We respectively injected a single dose of tamoxifen (100 mg/kg, i.p.) into 2 weeks, 4 weeks, and 8 weeks old *Prrx1^CreER*; *Rosa26^tdTomato* mice and performed immunostaining at 12 weeks (***Figure 3d***). We found that most of the cells in the enthesis originated from *Prrx1*+ cells at the 2 W-12W group. At the same time, this involvement decreased significantly at the 4 W-12W group and disappeared at the 8 W-12W group. *Prrx1*+ cells participated in the development of enthesis, including the continuous four gradient layer structure: bone, calcified fibrocartilage, uncalcified fibrocartilage, and tendon (***Figure 3e and f***). We have isolated the *Prrx1*+ cells and verified that it has the stem-like cell phenotype (***Figure 3—figure supplement 1***). These finding suggested that *Prrx1*+ cell was a vital subpopulation of mesenchymal stem cells for enthesis development.

To verify whether *Prrx1*+ cells participated in adult murine enthesis injury healing, *Prrx1^CreER*; *Rosa26^tdTomato* mice (12 weeks old) were performed RC injury after injected a single dose of tamoxifen (100 mg/kg, i.p.). At postoperative 4 weeks, mice were sacrificed for immunofluorescence (***Figure 1g***). We found that *Prrx1*+ cells were activated and migrated from the surrounding area to the injury site to participate in the enthesis healing via differentiating into osteocytes or chondrocytes (***Figure 3h and i***).

## Proper mechanical stimulation improves the enthesis injury repair

To find out if the proper mechanical stimulation could improve enthesis injury repair, the mice began to receive treadmill training at 1 week after enthesis surgery with different treadmill training (0 min per day, 10 min per day, 20 min per day, and 30 min per day, 5 consecutive days per week). At 4 and 8 weeks after surgery, mice were sacrificed for histology and mechanical test analysis (***Figure 4a***). We found that treadmill training with 20 min per day showed better tissue maturation, collagen arrangement (***Figure 4b***), higher histological scores (***Figure 4c***), and more fibrocartilage regeneration (***Figure 4d***). The best mechanical results of RC have also occurred at the group receiving 20 min treadmill training per day (***Figure 4e***). These results indicated that proper mechanical stimulation could improve enthesis healing.

## Proper mechanical stimulation mobilize the *Prrx1*+ cells to participate in enthesis injury repair

To investigate the potential role of mechanical stimuli on *Prrx1*+ cells and the relationship between *Prrx1*+ cells number and repair quality, we performed lineage tracing analysis using *Prrx1^CreER*; *Rosa26^tdTomato* mice. After receiving enthesis injury repair surgery followed with a single dose of tamoxifen (100 mg/kg, i.p.), the mice started to receive different treadmill training (0 min per day, 10 min per day, 20 min per day, and 30 min per day, 5 consecutive days per week) at 1 week after surgery (***Figure 5a***). We found that *Prrx1*+ cells were absent at the enthesis in adult mice (***Figure 5b***). *Prrx1*+ cells could migrate from the nearby tissue to the healing area at 2 weeks after surgery (***Figure 5b***). The 10 min and 20 min treadmill training could significantly enhance the migration of *Prrx1*+ cells to the healing area compared with the group without treadmill training. Excessive treadmill training decreased the migration of *Prrx1*+ cells to the healing area (***Figure 5c***).

## TGF-β1 mediated mechanical stimulation to enhance enthesis injury repair

Previous report showed that TGF-β1 can recruit mesenchymal stem cells to maintain the balance of bone resorption and formation (***Tang et al., 2009***). The GO analysis found that *Prrx1*+ cells were highly

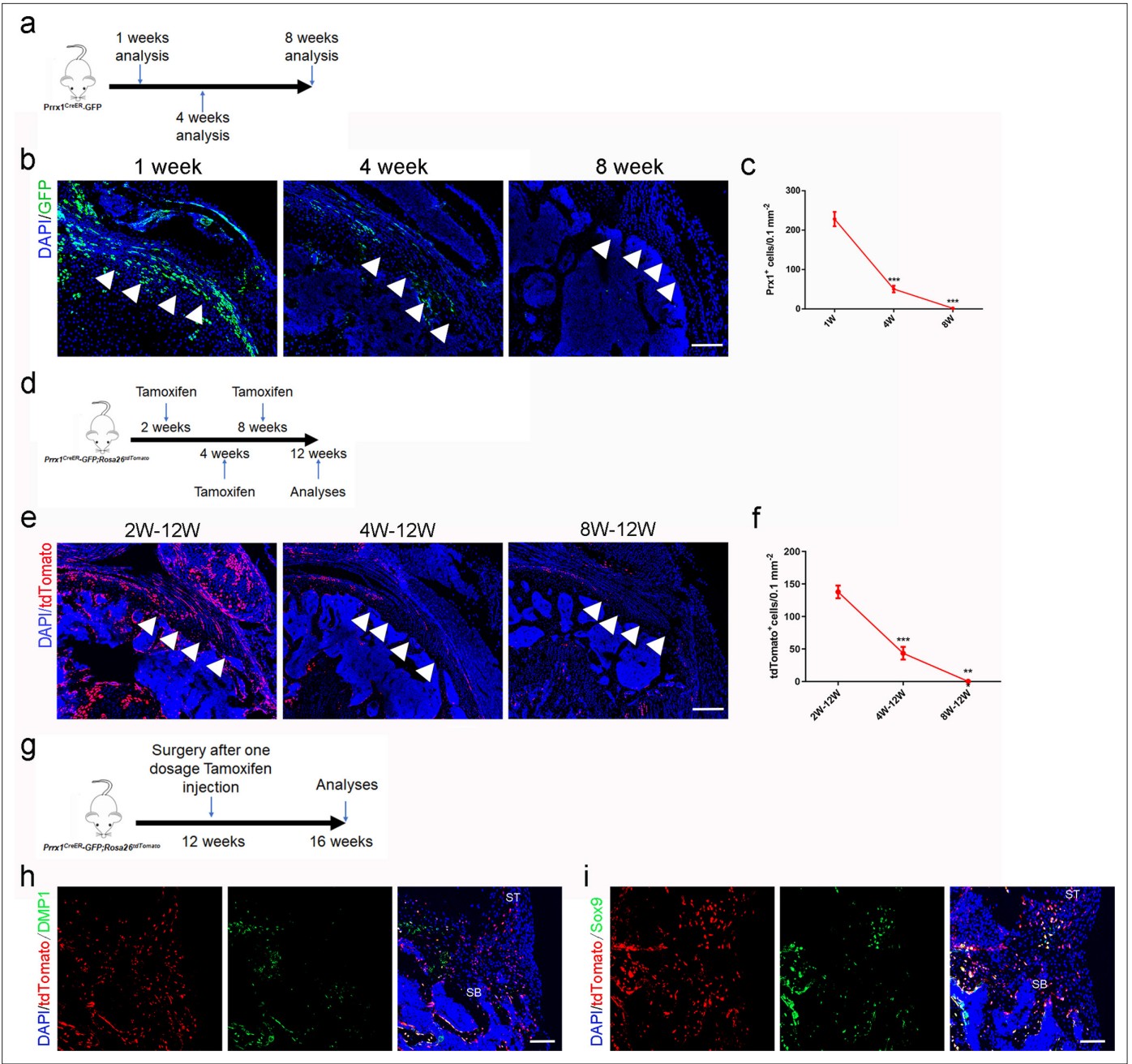

**Figure 3.** *Prrx1+* cells are involved in rotator cuff enthesis development and injury regeneration. (**a**) Schematic diagram of *Prrx1^CreER-GFP* mice, which were sacrificed at 1, 4, and 8 weeks after surgery for immunofluorescent analysis. (**b**) Representative immunofluorescent images of GFP staining of active *Prrx1+* cells (green) and DAPI (blue) staining of nuclei in murine humeral head at postnatal 1, 4, and 8 weeks. Scale bars, 200 μm. (**c**) Quantitative analysis of the number of active *Prrx1+* cells in enthesis. n = 6 per group. (**d**) Schematic diagram of *Prrx1^CreER-GFP*; *Rosa26^tdTomato* mice, which were sacrificed for immunofluorescent analysis at 12 weeks after tamoxifen administration at Postnatal 2, 4, and 8 weeks. (**e**) Representative immunofluorescent images of tdTomato+ cells (*Prrx1+* cells, red) and DAPI (blue) staining of nuclei in murine humeral head at postnatal 12 weeks after injection with tamoxifen respectively at postnatal 2, 4, and 8 weeks. Scale bars, 200 μm. (**f**) Quantitative analysis of the number of tdTomato+ cells in enthesis. n = 6 per group. (**g**) Schematic diagram of *Prrx1^CreER-GFP*; *Rosa26^tdTomato* mice which were received acute enthesis injury and sacrificed for immunofluorescent analysis at 4 weeks after surgery after sigle dose tamoxifen injection. (**h**) Representative immunofluorescent images of tdTomato+ cells (*Prrx1+* cells, red) in murine enthesis at postoperative 4 weeks. Scale bars, 100 μm. (**i**) Quantitative analysis of the number of tdTomato+ cells in enthesis. n = 6 per group. All data were reported as mean ± SD. The white triangles indicated the area of enthesis. SB, subchondral bone; ST: supraspinatus tendon. *p < 0.05, **p < 0.01, ***p < 0.001.

The online version of this article includes the following source data and figure supplement(s) for figure 3:

*Figure 3 continued on next page*

*Figure 3 continued*

**Source data 1.** The source data of quantitative analysis of the number of active *Prrx1*⁺ cells in enthesis for **Figure 3c** and quantitative analysis of the number of tdTomato⁺ cells in enthesis for **Figure 3f**.

**Figure supplement 1.** *Prrx1*⁺ cells and verified that it has the stem-like cell phenotype.

responded to TGF-β at P7, when enthesis initial mineralization began. Therefore, we hypothesized that TGF-β1 played an important role in enthesis repair process. First, we harvested the enthesis samples and nearby humeral head perichondrium to perform ELISA analysis to reveal the content of active TGF-β1 during enthesis repair procedure (**Figure 6a**). We found that active TGF-β1 concentration increased and reached the peak at 2 weeks after surgery. Then, it returned to its basal level at 10 weeks. Mechanical stimulation could stimulate the release of active TGF-β1 during the repair procedure. Meantime, there is an active TGF-β1 concentration gradient between enthesis and nearby perichondrium (**Figure 6b**).

To investigate the role of TGF-β1 during enthesis repair procedure, mice were recieved enthesis surgery and treated with or without TGF-β1 neutralizing antibody (ab64715). At 4 and 8 weeks, mice were sacrificed for histological and mechanical test analysis. The results showed that highly cellular, fibrovascular granulation tissue were observed at the supraspinatus enthesis in all groups at 4 weeks after surgery. Fibrovascular scar in the groups without ab64715 were relatively organized in H&E staining. Results of toluidine blue/fast green staining exhibited that mature fibrocartilage occurred more in groups without ab64715 than other groups with ab64715 (p < 0.05 for all). Well-organized soft tissue and tidemark at the enthesis occurred at 8 weeks after surgery. However, enthesis in groups treated with ab64715 showed weak remodeling tissue than the groups without ab64715. The fibrocartilage was thinner in the groups with ab64715 than that in other groups without ab64715 (p < 0.05 for all) (**Figure 6c and d**). The mechanical test showed that groups with ab64715 had lower failure load and stiffness than other groups without ab64715 at each time point (p < 0.05 for all) (**Figure 6e**).

To understand if TGF-β1 also mediated the migration of *Prrx1*⁺ cells to participate in enthesis injury repair, we performed lineage tracing analysis using *Prrx1^CreER^; Rosa26^tdTomato^* mice. After a single dose of tamoxifen (100 mg/kg, i.p.) injection, the mice were received surgery to create a enthesis repair model with or without ab64715 treatment. Mice began to receive treadmill training (20 min per day, 5 consecutive days per week) at 1 week after surgery and were sacrificed for immunofluorescence analysis at 2 weeks. We found that treadmill training could enhance *Prrx1*⁺ cells to the healing area and this effect could be eliminated by the treatment with TGF-β1 neutralizing antibody (**Figure 6f**).

To find out if mechanical stimulation could have indipendent effect on *Prrx1*⁺ cells migration, we isolated *Prrx1*⁺ cells and investigated the effect of mechanical stimulation on *Prrx1*⁺ cells with or without TGF-β1. We used a special dish that could load tensile force to *Prrx1*⁺ cells (**Figure 6—figure supplement 1a**). After 4 consecutive days of mechanical stimuli (5%, 0.5 Hz, 20 min per day) with TGF-β1 (0.4 ng/ml), *Prrx1*⁺ cell migration ability was analyzed by scratch assay and Transwell assay. We found that mechanical stimulation could not indipendently enhance the *Prrx1*⁺ cell migration ability. At the same time, TGF-β1 could improve its migration ability, and this effect could be significantly stimulated by mechanical stimulation (**Figure 6—figure supplement 1b, c, d**). Western blot showed that pSmad2/3 was activated during this process (**Figure 6—figure supplement 1e**, f). These results indicated that treadmill training mobilised *Prrx1*⁺ cells to enhance enthesis injury repair mainly by mediating the release of active TGF-β1.

## Primary cilia was essential for TGF-β signaling to promote enthesis injury repair

Previous studies showed that there were many receptors in primary cilia, which played an essential role in signal transmission ( *Anvarian et al., 2019*; *Villalobos et al., 2019*; *Dalbay et al., 2015*; *Pala et al., 2017*). To determine whether the primary cilia plays an essential role in the transmission of TGF-β signaling, we created the primary cilia conditional knocked out transgenic mice. *Prrx1^CreER^; Ift88^flox/flox^; Rosa26^tdTomato^* mice and *Prrx1^CreER^; Rosa26^tdTomato^* mice were received enthesis injury repair surgery after 5 days continuous tamoxifen injection (75 mg/kg, i.p.). The mice were recieved treadmill training at the day 7 after surgery (20 minitues per day, 5 days per week) and sacrificed for assessment at 4 and 8 weeks. Results showed that conditional ablation of *Ift88* in *Prrx1*⁺ cells significantly damaged the

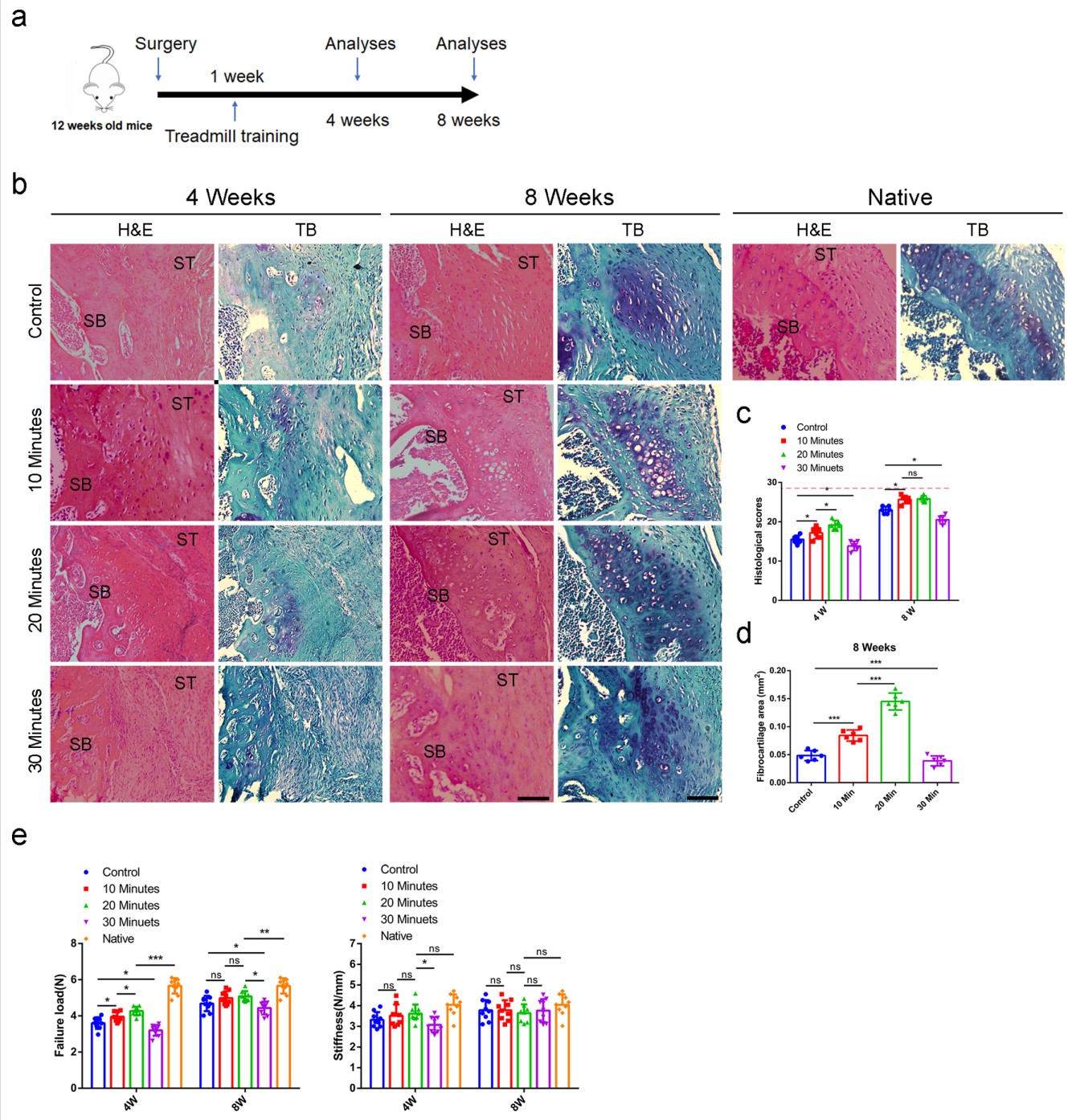

**Figure 4.** Proper mechanical stimulation could improve the enthesis injury repair. (**a**) Schematic diagram of mice, which were received enthesis surgery and sacrificed for immunofluorescent analysis at 4 and 8 weeks after surgery. (**b**) Representative image of H&E and Toluidine blue/Fast green staining of enthesis. Scale bar, 200 μm. (**c**) Quantitative analysis of H&E score. The red dolt line indicated the perfect histological score of 28. n = 6 per group. (**d**) Quantitative analysis of fibrocartilage thickness. n = 6 per group. (**e**) Quantitative analysis of Failure Load and stiffness. n = 9 per group. SB, subchondral bone; ST: supraspinatus tendon. Data were presented as mean ± SD. *p < 0.05, **p < 0.01, ***p < 0.001, ns p > 0.05.

The online version of this article includes the following source data for figure 4:

**Source data 1.** The source data of quantitative analysis of H&E score for *Figure 4c*.

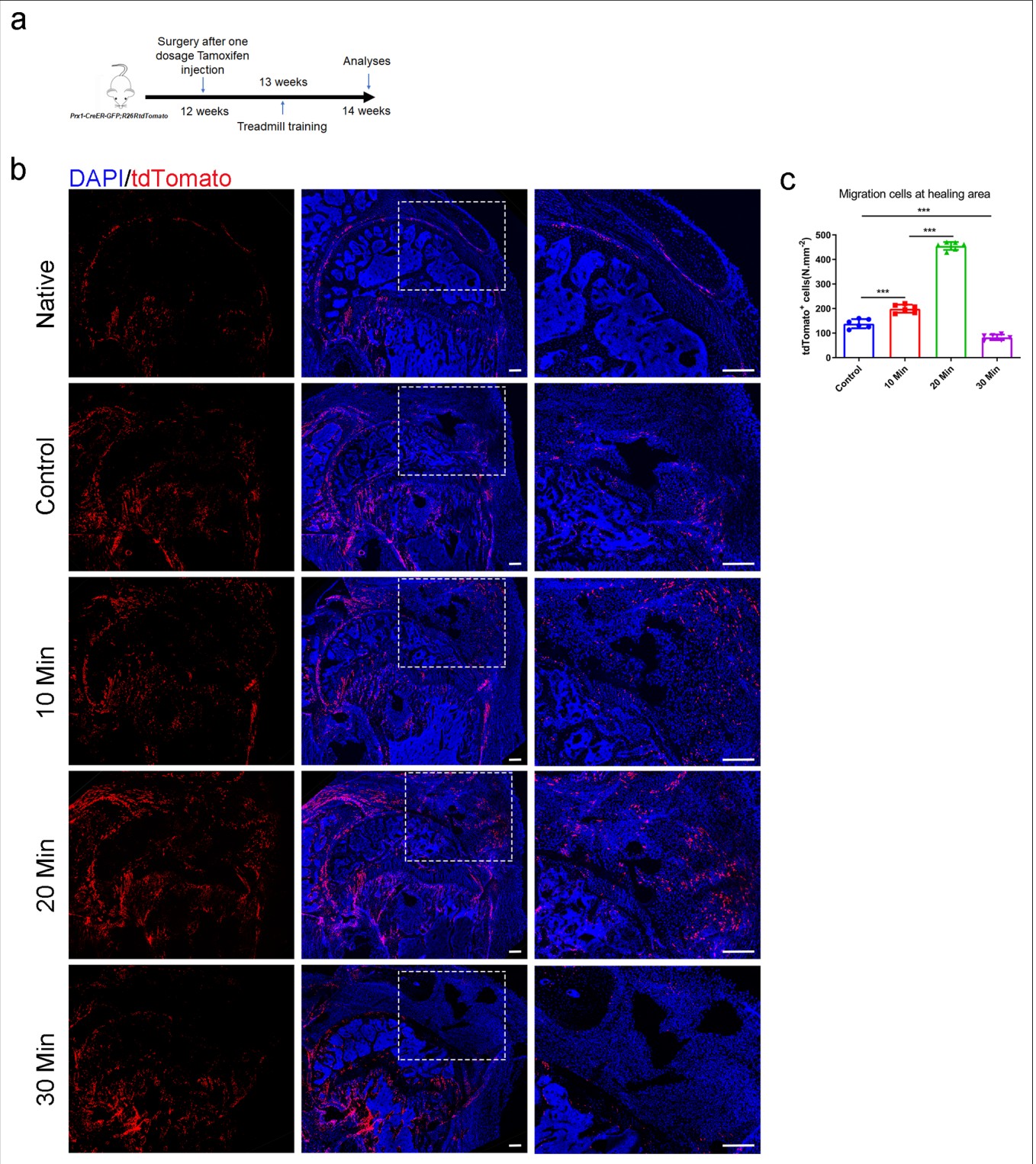

**Figure 5.** Proper mechanical stimuli could enhance the migration of *Prrx1*+ cells to participate in enthesis healing. (**a**) Schematic diagram of *Prrx1*^CreER-GFP^; *Rosa26*^tdTomato^ mice which were received enthesis surgery and sacrificed for immunofluorescent analysis at 14 weeks after surgery after sigle dose tamoxifen injection. (**b**) Representative immunofluorescent images of Tdtomato (red) staining of *Prrx1*+ cells and DAPI (blue) staining of nuclei under different mechanical stimuli. Scale bar, 200 µm. (**c**) Quantitative analysis of migration *Prrx1*+ cells at the healing area. n = 6 per group. Data were presented as mean ± SD. Scale bar, 200 µm. *p < 0.05, **p < 0.01, ***p < 0.001.

The online version of this article includes the following source data for figure 5:

**Source data 1.** The source data of quantitative analysis of migration *Prrx1*+ cells at the healing area for *Figure 5c*.

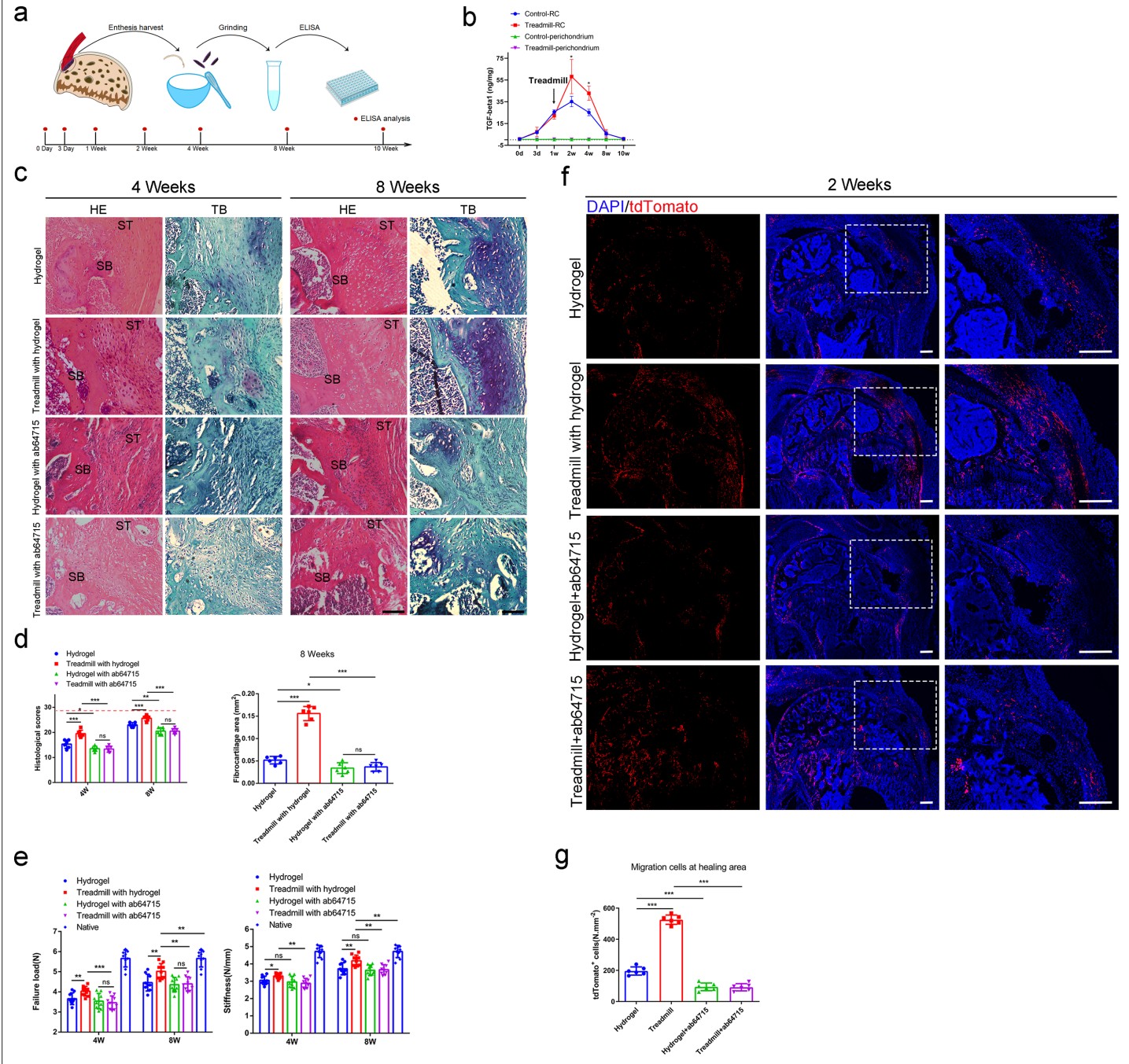

**Figure 6.** TGF-β1-mediated mechanical stimulation to enhance enthesis injury repair. (**a**) Schematic diagram of ELISA analysis. (**b**) ELISA analysis of TGF-β1 concentration during the enthesis healing procedure. n = 3 per group. (**c**) Representative image of H&E and Toluidine blue/Fast green staining of enthesis. Scale bar, 200 µm. (**d**) Quantitative analysis of H&E score and fibrocartilage thickness. The red dolt line indicated the perfect histological score of 28. n = 6 per group. (**e**) Quantitative analysis of Failure Load and stiffness. n = 9 per group. (**f**) Representative immunofluorescent images of Tdtomato (red) staining of *Prrx1*+ cells and DAPI (blue) staining of nuclei under different mechanical stimuli at postoperative 2 weeks. Scale bar, 200 µm. (**g**) Quantitative analysis of migration *Prrx1*+ cells at the healing area. n = 6 per group. Data were presented as mean ± SD. *p < 0.05, **p < 0.01, ***p < 0.001.

The online version of this article includes the following source data and figure supplement(s) for figure 6:

**Source data 1.** The source data of TGF-β1 concentration during the enthesis healing procedure for *Figure 6b*.

**Figure supplement 1.** Mechanical stimulation could amplify the transmission of TGF-β signaling.

**Figure supplement 1—source data 1.** The source data of quantitative analysis of migration cells in a scratch assay for *Figure 6—figure supplement*

*Figure 6 continued on next page*

Figure 6 continued

1c.

**Figure supplement 1—source data 2.** The the original files of the full raw unedited gels.

primary cilia (*Figure 7a and b*). Without the primary cilia, mechanical stimulation could not enhance the migration of *Prrx1*+ cells to the healing area (*Figure 7c and d*). Results of H&E staining showed that less scar tissue formed at the enthesis at 4 weeks after surgery in the three groups, and there was no significant difference in histological scores between the primary cilia dysfunction groups with or without treadmill training. No fibrocartilage was found in both groups at this time point. At 8 weeks after surgery, H&E staining showed high cellular, fibrovascular granulation tissue at the enthesis. Meanwhile, few fibrocartilage tissues were found at the enthesis site in these two *Ift88* damaged groups. There was no significant difference in H&E scores and the fibrocartilage area between the mice with or without treadmill training (*Figure 7e, f and g*). No significant difference in failure load and stiffness was found between the mice with or without treadmill training (*Figure 7j*).

## TGF-β1 enhanced the migration of *Prrx1*+ cells via ciliary TGF-β signaling

To investigate the relationship between primary cilia and TGF-β signaling pathway, we isolated the *Prrx1*+ cells and examined the distribution of TGF-β receptor 2 (TGF-βR2) in the cells with or without mechanical stimulation. Results showed that TGF-βR2 existed on the surface of *Prrx1*+ cells. At the same time, TGF-βR2 was concentrated in the primary cilia under the effect of TGF-β1 (0.4 ng/ml), and mechanical stimuli could improve TGF-βR2 translocated into the primary cilia (*Figure 8a and b*). To understand if ciliary TGF-βR2 was essential for TGF-β signaling transmission, we used shRNA to knock down *Pallidin* in *Prrx1*+ cells, which could inhibit TGF-βR2 translocating into the primary cilia (*Zheng et al., 2018*). At the same time, *Ift*88 was knock out by lentivirus expressing Cre in vitro (*Figure 8—figure supplement 1*). Results showed that *Pallidin* in *Prrx1*+ cells could significantly be knocked down by shRNA (*Figure 8c and d*). TGF-βR2 concentrating in primary cilia was decreasing markedly in *Pallidin* knocked down group (*Figure 8e and f*). The results of scratch assay and transwell assay showed that the effect of TGF-β1 on *Prrx1*+ cells migration ability was eliminated in *Pallidin* knocked down and No cilia group (*Figure 8g, h and i*). Western blot analysis showed that the Smad2/3 signaling pathway was also inhibited at the same time (*Figure 8j and k*).

## Discussion

*Prrx1*+ cells and mechanical stimulation have been extensively studied during the skeletal development (*Yuan et al., 2015*; *Yuan et al., 2016*). However, their role in enthesis regeneration is poorly understood. In this study, we identified that *Prrx1*+ cells are involved in enthesis development, but may not be important player in adult enthesis. However, during injury repair, *Prrx1*+ cells could migrate to the injury area to participate in enthesis healing. Proper mechanical stimulation could increase the release of TGF-β1 to immobilize *Prrx1*+ cells and promote enthesis injury repair. Ciliary TGF-βR2 was essential for TGF-β signaling transmission during proper mechanical stimulation promoting enthesis injury repair procedure. As far as we known, this is the first report uncovering the characteristics of *Prrx1*+ cells on enthesis development and injury healing, and provided new insights into the progenitor source for enthesis injury repair. Meanwhile, this study found a new mechanism about the mechanical stimulation signal transmission, and had pieced together a mechanical conduction mechanism.

The microstructure regeneration of enthesis is a difficult issue considering the complicated structure of enthesis consisting of four continuous gradient layers: bone, calcified fibrocartilage, uncalcified fibrocartilage and tendon, and low self-regeneration ability (*Chen et al., 2019*). Therefore, current treatments aim to regenerate enthesis microstructure to acquire reliable long-term clinical results. A previous study found that low-intensity pulsed ultrasound stimulation after autologous adipose-derived stromal cell transplantation could improve the fibrocartilage and bone regeneration, leading to a better enthesis healing quality (*Lu et al., 2016*). Recently, tissue engineering was prevalent for repairing enthesis injury and showed promising results of fibrocartilage and bone regeneration, associated with better mechanical testing results (*Chen et al., 2019*; *Chen et al., 2020*; *Tang et al., 2020*).

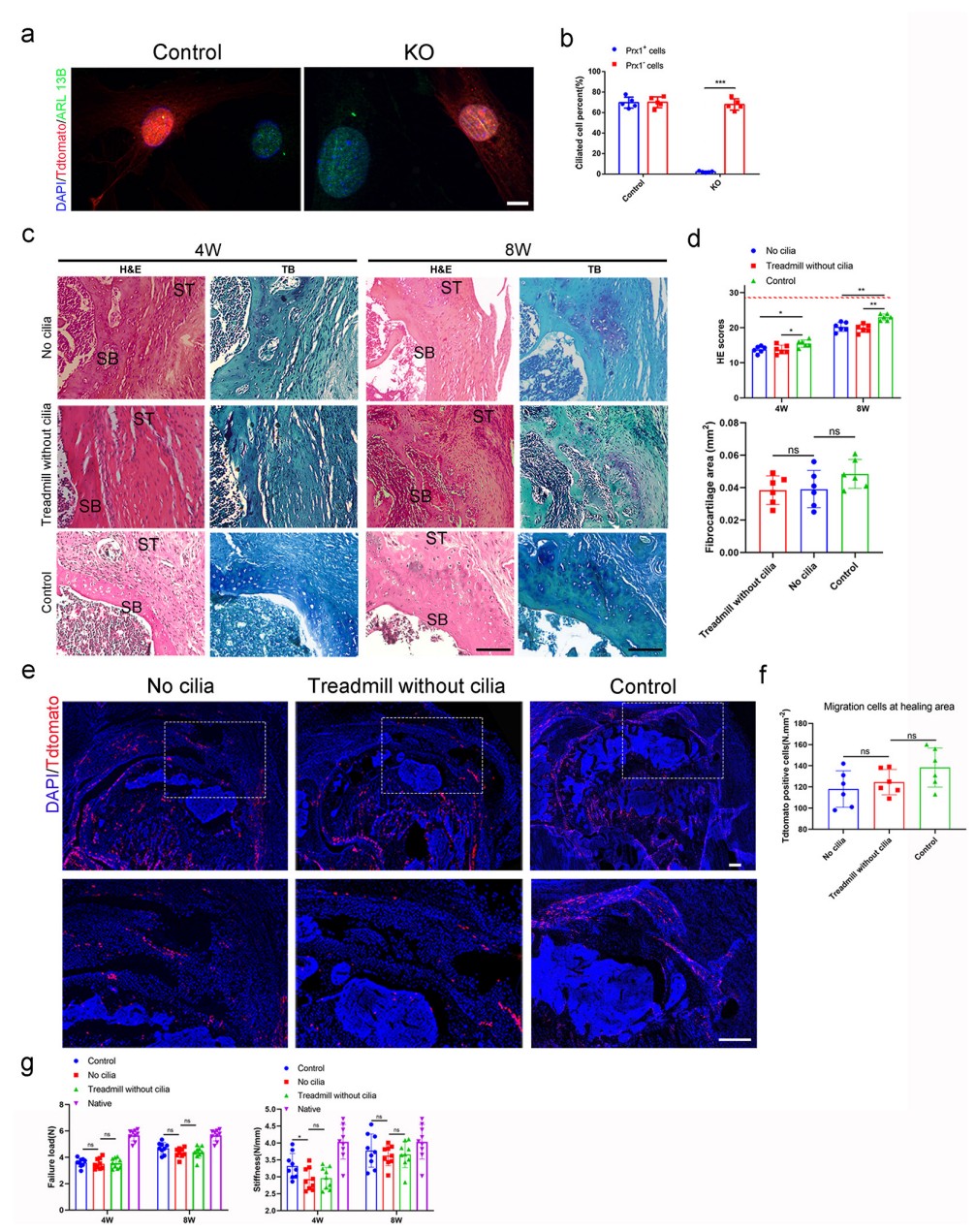

**Figure 7.** Primary cilia was essential for TGF-β signaling to promote enthesis injury repair. (**a**) Representative immunofluorescence image of tdTamato (red) staining of *Prrx1*+ cells, ARL 13B (green) staining of primary cilia, and DAPI (blue) staining of nuclei. Scale bar, 5 μm. (**b**) Quantitative analysis of ciliated cell percent in *Prrx1*+ cells and *Prrx1*- cells. n = 5 per group. (**c**) Representative image of H&E and Toluidine blue/Fast green staining of enthesis. Scale bar, 200 μm. (**d**) Quantitative analysis of H&E score and fibrocartilage area at the enthesis. The red dotted line stands for perfect H&E scores of 28. n = 6 per group. (**e**) Representative immunofluorescence image of tdTamato (red) staining of *Prrx1*+ cells, DAPI (blue) staining of nuclei at the enthesis. Scale bar, 200 μm. (**f**) Quantitative analysis of Tdtomato+ cells in the healing area. n = 6 per group. (**g**) Quantitative analysis of load failure and stiffness of enthesis. n = 9 per group. SB, subchondral bone; ST: supraspinatus tendon. Data were presented as mean ± SD. ***p < 0.001.

The online version of this article includes the following source data for figure 7:

**Source data 1.** The source data of quantitative analysis of ciliated cell percent in *Prrx1*+ cells and *Prrx1*- cells for *Figure 7b*.

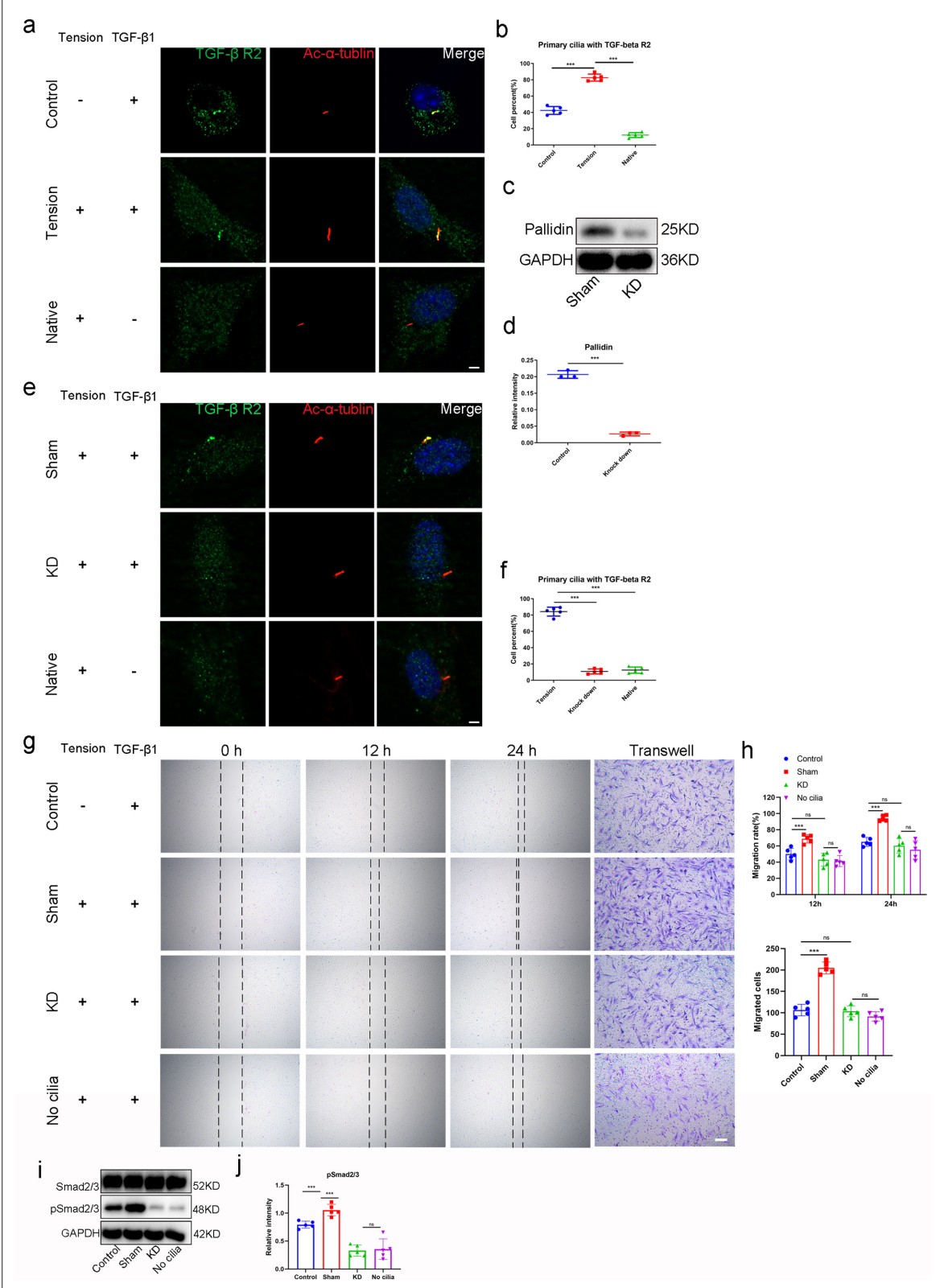

**Figure 8.** TGF-β1 enhanced the migration of *Prrx1*+ cells via ciliary TGF-β signaling. (**a**) Representative image of immunofluorescent analysis of TGF-βR2 (green), Ac-α-tubulin (red) staining of primary cilia, and DAPI (blue) staining of nuclei stimulation by mechanical force with TGF-β1 (0.4 ng/ml) in *Prrx1*+ cells. Scale bar, 5 μm. n = 5 per group. (**b**) Quantitative analysis of cell percent that the TGF-βR2 was concentrated in the primary cilia. n = 5 per group. (**c**) Western blot analysis of Pallidin in groups treated with or without *Pallidin*-siRNA. (**d**) Quantitative analysis of western blot. n = 5 per group.

*Figure 8 continued on next page*

*Figure 8 continued*

(**e**) Representative image of immunofluorescent analysis of TGF-βR2 (green), Ac-α-tubulin (red) staining, and DAPI (blue) staining of nuclei stimulation by mechanical force with TGF-β1 (0.4 ng/ml) in *Prrx1*⁺ cells treated with or without *Pallidin*-siRNA. Scale bar, 5 μm. (**f**) Quantitative analysis of cell percent that the TGF-βR2 was concentrated in the primary cilia. n = 5 per group. (**g**) Scratch assay and transwell assay of *Prrx1*⁺ cells. Scale bar, 100 μm. (**h**) Quantitative analysis of scratch assay. n = 5 per group. (**i**) Western blot analysis of Smad2/3/pSmad2/3 signaling. (**j**) Quantitative analysis of western blot. n = 5 per group. Data were presented as mean ± SD. *p < 0.05, **p < 0.01, ***p < 0.001.

The online version of this article includes the following source data and figure supplement(s) for figure 8:

**Source data 1.** The source data of quantitative analysis of cell percent that the TGF-βR2 was concentrated in the primary cilia for *Figure 8b*.

**Source data 2.** The the original files of the full raw unedited gels.

**Figure supplement 1.** Knock out Ift88 in vitro.

**Figure supplement 1—source data 1.** The source data of quantitative analysis of western blot for *Figure 8b* The source data of quantitative analysis of cell percent that the ciliated cells for *Figure 8d*.

**Figure supplement 1—source data 2.** The the original files of the full raw unedited gels.

In the present studies, we found that better regeneration of fibrocartilage was higly correlated with the number of *Prrx1*⁺ cells and was accompanied by better mechanical testing results, which was consistent with previous reports.These data suggested that the fibrocartilage regeneration directly influenced the enthesis healing quality and could be a reliable indicator for the enthesis regeneration.

Mechanical stimulation is a common therapy in the clinic, while it is a double-edged sword for enthesis healing. On one hand, appropriate mechanical load could stimulate local blood circulation, promote mesenchymal stem cell differentiation, and help releasing anti-inflammatory factors to improve tissue healing (*Olesen et al., 2006*; *Jonsson et al., 2008*; *Ohberg et al., 2004*). *Wang et al., 2017* reported if the training started early, such as 96 hr after acute patellar tendon enthesis injury, it will lead to better recovery results with fibrocartilage and mechanical test parameters in a rabbit model. On the other hand, excessive mechanical loading in the early healing phase might delay or even inhibit tissue healing (*Lee et al., 2012*). Rodeo et al found that immediate excessive treadmill training at the early stage causes delayed enthesis healing in the murine enthesis repair model (*Wada et al., 2019*). These reports suggested that enthesis could only respond favorably to controlled loading after injury. In the present studies, we chose a 7 days delayed mechanical stimulation and tested three sets of mechanical intensity. We found that 20 min per day mechanical stimulation (7 days delayed, 10 m/min, 5 days per week) could enhance *Prrx1*⁺ cell migration, improve the quality of fibrocartilage and bone regeneration in the enthesis, resulting in better biomechanical results. Our finding suggested that appropriate mechanical stimulation (7 days delayed, 10 m/min, 20 min per day, 5 days per week) indeed improve early enthesis healing, and this model was reliable to uncover the mechanism of mechanical stimulation-induced enthesis healing. Still, we should notice that we didn't figure out the best mechanicla stimulation intensity here, which could be further explored in future studies.

Our results showed that proper mechanical stimuli could enhance the migration of *Prrx1*⁺ cells and *Prrx1*⁺ cells could differentiate in cartilage and bone cells which play a critical role in enthesis healing. We know that *Prrx1*⁺ cells are important in skeletal development, especially during the embryonic development (*Kawanami et al., 2009*). However, their role in enthesis development and contribution to skeletal tissue regeneration was poorly understood. In this study, we found that GFP⁺ cells (active *Prrx1*⁺ cells) were abundant in the enthesis during early stage and disappeared at adulthood. The tdtomato⁺ cells mainly localized within the periosteum, perichondrium, and growth plate at adulthood, which indicated that *Prrx1*⁺ cells confined at these places were active in young age and quiescence, but not completely disappear at adulthood. Meanwhile, there were no *Prrx1*⁺ cells at the enthesis area in adult mice. When enthesis injury happens, *Prrx1*⁺ cells could migrate to the injury area to participate in the healing process. Besides, our results showed that *Prrx1*⁺ cell numbers at the enthesis were related to the healing quality, suggesting that *Prrx1*⁺ cells were pivotal for enthesis healing. One limitation of this study is that we did not investigate other mesenchymal sub-population in the current study.

How did mechanical stimulation enhance the migration of *Prrx1*⁺ cells to the healing site? As we known, TGF-β signaling plays an essential role in tissue homeostasis and injury healing and TGF-β signaling activation is in precise spatial and temporal manner (*Zhen and Cao, 2014*; *Kim et al., 2018*; *Delaney et al., 2017*). The level of TGF-β1 was relatively high at the healing interface and TGF-β1

could promote the migration of MSCs to modulate bone remodeling (*Tang et al., 2009*). Robertson et al reported that TGF-β1 function is mainly regulated by its activation rather than synthesis or secretion (*Robertson and Rifkin, 2016*). Hence, we mainly focused on the activation of TGF-β1. Our results showed that TGF-β signaling is involved in enthesis healing. During this process, active TGF-β1 concentration was elevated, and was further enhanced by mechanical stimulation. In vitro,we found that mechanicla stimulation couldn't indipendently improve *Prrx1*+ cells migration ability, while it could enhance the sensitivity of the *Prrx1*+ cells to TGF-β1. Although we did not further reveal its effects on *Prrx1*+ cells migration in vivo, it still provided new insights in new mechanisms about the mechanical stimulation signal transmission. At the same time, we could not rule out the involvement of other signaling pathways in this process.

Primary cilia have been recognized as an essential cellular mechanoreceptor and mechanosensitive channels (*Bangs and Anderson, 2017*; *Hoey et al., 2012*). Still, the regulatory mechanism of primary cilia in mechanical stimulation transmission remains unclear, even controversial. Polycystin-1 (PC1) and polycystin-2 (PC2), co-distribution in the primary cilia of kidney epithelium, was reported to be involved in the regulation of intracellular $Ca^{2+}$ signaling and transfer mechanical stimuli (*Nauli et al., 2003*). In addition to PC1 and PC2, another primary cilia-based calcium channels-transient receptor potential (TRP) could also sense mechanical stimuli via conducting $Ca^{2+}$ signaling (*Luo et al., 2014*). Nonetheless, many calcium channels are not only localized in the ciliary membrane, but also other parts of the cells, so it is difficult to differentiate the difference between ciliary and cytosolic $Ca^{2+}$ in response to the same mechanical stimuli. Delling et al found that cilia-specific $Ca^{2+}$ influxes were not observed in physiological or even highly supraphysiological levels of fluid flow (*Delling et al., 2016*). In this study, we found that knocking out Ift88 prevented the mechanotransduction. Inhibition of ciliary TGF-β signaling could decrease the mechanotransduction, suggesting that primary cilia could regulate mechanical stimuli via ciliary TGF-βR2. This finding provides new insights into the role of primary cilia in mechanical stimuli transmission. However, we didn't exclude ciliary $Ca^{2+}$ signaling or other ciliary signaling pathway participating in this mechanical stimulation transmission process in this study.

## Conclusion

In conclusion, *Prrx1*+ cells were an essential subpopulation of progenitors for enthesis development and injury repair. Mechanical stimulation could increase the release of TGF-β1 and enhance mobilization of *Prrx1*+ cells to promote enthesis injury repair via ciliary TGF-β signaling.

# Materials and methods
## Collection of single-cell suspension from Supraspinatus enthesis

In general, 10–12 suspensions tendon enthesis tissue (E15.5, P7 and P28), including the tendon (one millimeter in length) and the portion of the humeral head proximal to the growth plate near the tendon attachment, were collected from pooled sibling shoulders (five to six mice per pool). Fresh enthesis tissue were finely chopped with small scissors in 1 ml of Dulbecco's modified Eagle's medium (DMEM), then digested in 0.5% type I collagenase (Life Technologies) and 7 U/ml Dispase II (Gibco) at 37 °C for 30 min. Then the supernatant was collected and filtered through 70 µm cell filters (Falcon BD), and centrifuged for 5 min at 300 g, before re-suspending the pellet in DMEM containing 2% serum, and the cell suspension was kept on ice until load on chip.

## Flow cytometry and cell sorting

Collected cell suspension were blocked with purified rat anti-mouse CD16/CD32 (BD Pharmingen, dilution 1:100) for 10 min, then stained with fluorophore conjugated antibodies. Antibodies used in this study are anti-mouse CD45-APCCy7, Ter119-APCCy7 (Biolengend), DAPI (eBiosciences) stain was used to exclude dead cells. For cell sorting, single cells were gated using doublet-discrimination parameters and cells were collected in FACS buffer (1 x HBSS, 2% FBS, 1 mM EDTA). Cell viability was assessed with trypan blue and only samples with >85% viability were processed for further sequencing.

## scRNA-Seq sequencing, data processing, and quality control

Around 10,000 sorted live CD45-Ter119- cells for each timepoint sample were resuspended in FACS buffer according to the recommendations provide by 10 × Genomics for optimal cell recovery.

Single-cell mRNA libraries were built using the Chromium Single Cell 3′ kit (v3), libraries sequenced on an Illumina NovaSeq 500 instrument. Single-cell fastq sequencing reads from each sample were processed by aligning reads and obtaining unique molecular identifier (UMI) counts and converted to gene expression matrices, after mapping to the mouse (mm10) reference genome using the Cell Ranger v4.0.0 pipeline, according to the standard workflow (10 × Genomics).

## scRNA-seq data ananlysis

Quality control was conducted for each dataset, cells with less than 200 genes and the top 10% cells were removed to minimize multiplet possibility. Cells were retained if the percent mitochondrial reads were lower that 20% (8919 cells for embryonic day 15.5, 7489 cells for postnatal day 7 and 5124 cells for postnatal day 28). Data integration, graph-based cell clustering, dimensionality reduction, and data visualization were analyzed by the Seurat R package (v3.2). Data integration was performed via canonical correlation analysis to remove batch effect. Feature (gene) data was scaled in order to remove unwanted sources of variation using the Seurat ScaleData function for percent mitochondrial reads, number of genes detected and predicted cell cycle phase difference. Non-linear dimension reduction was performed using uniform manifold projection (UMAP) and graph-based clustering was performed using the Louvain algorithm. The number of statistically significant principal components were set empirically by testing top 10 differentially expressed genes (MAST method) between the clusters. Subsetting was performed by assessing marker gene expression across clusters, Clusters associated with the following cell-types were excluded from our analysis: muscle cells, immune cells blood cells, endothelial cells and undefined cells rich of histone genes. Functional annotation of the marker genes relative to GO terms was performed using ClusterProfiler (v3.18). Trajectory analysis of the tendon enthesis development was performed using Monocle (v2.4.0).

## Animals and treatment

The $Prrx1^{CreER-GFP}$ (Strain origin: C57BL6N/129; Stock No: 029211), $Ift88^{flox/flox}$ (Strain origin: C57BL6N/129; Stock No: 022409); $Rosa26^{tdTomato}$ (Strain origin: C57BL6N/129; Stock No: 007909) mouse strain was purchased from Jackson Laboratory (Bar Harbor, ME).

$Prrx1^{CreER-GFP}$ mice were crossed with $Ift88^{flox/flox}$ mice. The offspring were intercrossed to generate the following genotypes: WT, $Prrx1^{CreER-GFP}$, $Prrx1^{CreER-GFP}$; $ITF88^{flox/flox}$. Then, $Prrx1^{CreER-GFP}$; $ITF88^{flox/flox}$ mice were crossed with $Rosa26^{tdTomato}$ mice. The offspring were intercrossed to generate the following genotypes: $Prrx1^{CreER-GFP}$; $Rosa26^{tdTomato}$ mice (mice expressing tdTomato driven by Cre recombinase in $Prrx1^+$ cells), $Prrx1^{CreER-GFP}$; $ITF88^{flox/flox}$; $Rosa26^{tdTomato}$ mice (conditional deletion of $ITF88$ in $Prrx1$ lineage cells and expressing tdTomato, referred to as $Ift88^{-/-}$ herein). To induce Cre recombinase activity, we injected mice at designated time points with tamoxifen (75 mg/kg, i.p.) for consecutive five days. Since $Prrx1^{CreER-GFP}$ mice can express GFP in $Prrx1^+$ cells, we used $Prrx1^{CreER}$-GFP mice to real-time lable $Prrx1^+$ cells and $Prrx1^{CreER-GFP}$; $Rosa26^{tdTomato}$ mice to permenantly lable $Prrx1^+$ cells.

The genotype of the mice was determined by PCR analysis of genomic DNA, extracted from mouse tails using the primers as follows. $Prrx1^{CreER-GFP}$ allele forward, 5'- ATACCGGAGATCATGCAAGC -3', reverse, 5'-GGCCAGGCTGTTCTT CTTAG-3', control forward, 5'- CTAGGCCACAGAATTGAAAG ATCT-3' and control reverse, 5'- GTAGGTGGAAATTCTAGCATCATCC-3'; $Ift88^{-/-}$ allele forward, 5'-TGAGGACGACCTTTACTCTGG-3′, and reverse, 5'-CTGCCATGACTGGTTCT CACT-3'; $Rosa26^{tdTomato}$ allele forward, 5'- AAGGGAGCTGCAGTGGAGTA-3', reverse, 5'-CCGAAAATCTGTGGGAAGTC-3', control forward, 5'-GGCATTAAAG CAGCGTATCC-3' and control reverse, 5'- CTGTTCCTGTACGGCA TGG-3'.

## Rotator cuff injury repair model

Twelve weeks old male mice underwent rotator cuff injury repair using protocol as previously reported (*Zhang et al., 2021*; *Bell et al., 2015*). After anesthetized with pentobarbital (0.6 mL/20 g; Sigma-Aldrich, St. Louis, MO), a longitudinal skin incision was made to expose the deltoid muscle of the left shoulder, and a transverse cut was made on it. The acromion was pulled away to expose the supraspinatus tendon. After the supraspinatus tendon was grasped with 6–0 Prolene (Ethicon, Somerville, NJ, USA), it was sharply transected at the insertion site on the greater tuberosity, and fibrocartilage layer was removed with a scalpel blade. A bone tunnel was made transversely to the distal greater tuberosity. Then, the suture was passed through the drilled hole and tied the supraspinatus tendon to its

anatomic position. The skin and deltoid muscles were closed in layer. To block the TGF-β1, hydrogel loading with the TGF-β1 neutralizing antibody was used. Mice were allowed free cage activities. At postoperative 4 weeks and 8 weeks, the mice receiving treadmill exercise or not were sacrificed, and the supraspinatus-humeral head composites were harvested for further study.

## Mechanical load in vivo and in vitro

A motor-powered treadmill with 12 lanes was used to generate a mechanical load to the enthesis in vivo. Treadmill exercise was conducted as previously reported (*Zhang et al., 2021*; *Wada et al., 2019*). All the mice underwent 1-week adaptive training to get familiar with the lane environment before surgery. Treadmill speed was increased daily until all mice tolerated running at 10 m/min for 30 min per day. At postoperative day 7, the mice in the treadmill group ran at a speed of 10 m/min on a 0° declined lane for 10 min, 20 min, or 30 min, 5 days per week.

A cell load system (CellLoad-300, Hao Mian, China) was used to generate tensile mechanical load on *Prrx1+* cells. *Prrx1+* cells were seeded on a plate which could expand and contract under external forces, at a density of $1.5 \times 10^4$ /cm$^2$. The parameters were set as follows: 1 Hz, 5%, 20 min per day.

## Immunofluorescence

Humeral head and supraspinatus tendon composite were harvested and fixed in the 4% paraformaldehyde in PBS overnight at room temperature. After decalcified and dehydrated, samples were embedded in Tissue-Tek O.C.T. Compound (SAKURA, Torrance, USA) and cut into 10 μm thickness of sagittal sections. Cell samples were fixed in the 4% paraformaldehyde in PBS for 30 min at room temperature. Both the parts and cell samples were blocked in 5% BSA for 40 min at room temperature and incubated with the primary antibodies anti-DMP1 (Abcam, 1:400, ab13970), anti-GFP (Abcam, 1:400, ab13970 or 1:400, ab290), anti-Sox9 (Abcam, 1:400; ab185966), anti-TGF-βR2 (Abcam, 1:400, ab186838) at 4 °C overnight. After washing, the sections were then incubated with the respective secondary antibodies (1:500, Abcam) for 1 hr at room temperature and sealed with DAPI. The images were captured with a Leica TCS-SP8 confocal microscope (Leica, Germany).

## Histological analysis

After radiographic assay, fixed samples were decalcified in EDTA for 14 days, dehydrated in gradient ethanol, embedded in paraffin, and then cut into 5 μm slices. The sections were stained with hematoxylin and eosin for general histology analysis. Two blinded observers measured histological tendon maturing score according to a previous report Table S1(*Wada et al., 2019*).

## Biomechanical test

An Instron biomechanical testing system (Model 5942, Instron, MA) was used to detect the failure load and stiffness of these samples. The tendon was secured in a clamp using sandpaper, while the humerus firmly clamped with a vice grip. The specimens were tested at room temperature, and samples were preconditioned with 0.1 N and then loaded to failure at a rate of 0.1 mm/s. A consistent gauge length was used throughout the test. Data were excluded if the tendon slipped out of the grip or did not fail at the supraspinatus tendon attachment site.

## ELISA

The supraspinatus tendon insertion specimens and femeral head were harvested at postoperative 0 day, and 3 days, and 1, 2, 4, 8, and 10 weeks. We removed the muscle belly and kept the tendon and the portion of the humeral head proximal to the growth plate near the tendon attachment. The femeral head cavity was removed by a drill. Then, we determine the concentration of active TGF-β1 in the enthesis using the ELISA Development Kit (R&D Systems, Minneapolis, MN) according to the manufacturer's instructions.

## Cell culture

To obtain *Prrx1+* cells, 1–2 weeks old *Prrx1CreER-GFP* mice were sacrificed. The tibiae and femurs were dissected and excised into chips of approximately 1–3 mm$^3$ with scissors. Then, the chips were suspended into a 25 cm$^2$ plastic culture flask with 5 ml of α-MEM containing 10% (vol/vol) FBS in the presence of 3 mg/ml (wt/vol) of collagenase II (Sigma) and digested the chips for one h in a shaking

incubator at 37°C with a shaking speed of 150 rpm. Washed the enzyme-treated chips with α-MEM and got the *Prrx1*+ cells by FACS. Reseeded the cells and changed the medium every 48 hr. To verify the primary cilia could be dysfuncitoned in vivo, we isolated the *Prrx1*+ cells through *Prrx1*$^{CreER-GFP}$; *Ift88*$^{flox/flox}$ mice and *Prrx1*$^{CreER-GFP}$; *Ift88*$^{flox/flox}$; *Rosa26*$^{tdTomoto}$ mice after tamoxifen injection for 5 days (75 mg/kg, i.p.) in the same way.

## Lentiviral vector transfection

*Prrx1*+ cells were transfected with lentiviral vector, targeting *Pallidin* (shRNA#1: 5'-ATACACTGGAAC AAGAGATTT-3', shRNA#2: 5'-CGCCAAGCTGGTGACTAT AAG-3') for 12 hr using lentiviral vector (VectorBuilder, Cyagen Biosciences, Santa Clara, CA) at an MOI of 20. *Prrx1*+ cells were maintained in growth media for a further 72 hr before the application of a mechanical stimulation. To dysfunction primary cilia in vitro, *Prrx1*+ cells that were harvested from *Prrx1*$^{CreER-GFP}$; *Ift88*$^{flox/flox}$, were transfected with lentiviral vector (EF1α–3Flag-Cre-IRES-EGFP, Genechem, GCPL0185881), which could express Cre to knock out Ift88.

## Scratch assay

For scratch wound assay, *Prrx1*+ cells ($1.5 \times 10^4$ cells/cm$^2$) were seeded into a stretchable dish and cultured with tension load (5%, 1 Hz, 20 min per day) for three days. Cell monolayer could be formed at this time point. A straight scratch was produced using a pipette tip. After washed with PBS to remove floating cells, adherent, complete medium was added. Wound closure was imaged at the 0, 12, and 24 hr of incubation time points. The rate of wound closure was calculated as follows: Migration rate (%) = (A0 – An)/A0 ×100, where A0 represents the initial wound area, and An represents the remaining area of the wound at the appointed time.

## Transwell assay

After tension force load for 4 days, $1 \times 10^4$ *Prrx1*+ cells were resuspended in 100 μl α-MEM medium were loaded into the upper chamber of 24-well Transwell plate (Corning, NY, USA) with 8 μm pore-sized filters. Complete medium, which supplemented with containing 0.4 ng/ml TGF-β1, was added to the lower chamber. After 12 hr of incubation, cells that migrated to the lower surface of the filter were rinsed, fixed, and stained with 1% crystal violet. An optical microscope was used to photograph and count the migrated cells.

## Western blotting

*Prrx1*+ cells with or without tension force load were collected for extracting protein. Western blotting was performed with 10% sodium dodecyl sulfate-polyacrylamide gel electrophoresis. Then the proteins were transferred into a nitrocellulose membrane, and membranes were blocked by nonfat milk. After blocking, the nitrocellulose membranes were then incubated using anti- Pallidin (Proteintech, 1:500, 10891–2-AP), anti-Smad2/3 Ab (Abcam, 1:500, ab202445), anti-pSmad2/3 Ab (Abcam, 1:500, ab63399), anti-GAPDH (Proteintech, 1:1000, 110494–1-AP). The figures for western blotting were visualized using enhanced chemiluminescence reagent (Thermo Fisher Scientific, Waltham, USA) and imaged by the ChemiDoc XRS Plus luminescent image analyzer (Bio-Rad).

## Statistical analysis

The statistical results were analyzed by GraphPad Prism 7.0 software. Quantitative data were expressed as mean ± standard deviation (SD), and differences above 2 groups were evaluated using one-way ANOVA with post hoc test, while the histological scores was performed using the Mann-Whitney test. Statistical significance was set at <0.05.

## Study approval

All animal care protocols and experiments in this study were reviewed and approved by the Animal Care and Use Committees of the Laboratory Animal Research Center of our institute (No.201703222). All mice were maintained in the specific pathogen-free facility of the Laboratory Animal Research Center.

## Acknowledgements

This work was supported by the Key Program of National Natural Science Foundation of China (NO.81730068), the Science and Technology Major Project of Changsha (NO. kh2003015), the National Natural Science Foundation of China (81902192), and the Hunan Provincial Natural Science Foundation Project (2021JJ20093). Thanks to professor Xianghang Luo and Hui Xie for supporting of this research.

## Additional information

### Funding

| Funder | Grant reference number | Author |
| --- | --- | --- |
| National Natural Science Foundation of China | No. 81730068 | Hong Bin Lu |
| Major Science and technology progect of Changsha Science and Technology Bureau | No. 41965 | Hongbin Lu |
| National Natural Science Foundation of China | 81902192 | Can Chen |
| Hunan Provincial Natural Science Foundation Project | 2021JJ20093 | Can Chen |

The funders had no role in study design, data collection and interpretation, or the decision to submit the work for publication.

### Author contributions

Han Xiao, Data curation, Investigation, Methodology, Visualization, Writing – original draft; Tao Zhang, Methodology, Software, Validation; Changjun Li, Conceptualization, Methodology, Validation, Visualization; Yong Cao, Data curation, Formal analysis, Resources; Linfeng Wang, Formal analysis, Investigation, Methodology; Huabin Chen, Data curation, Investigation; Shengcan Li, Data curation, Investigation, Methodology; Changbiao Guan, Formal analysis, Investigation; Jianzhong Hu, Project administration, Visualization; Di Chen, Conceptualization, Methodology, Writing – review and editing; Can Chen, Conceptualization, Project administration, Resources, Supervision, Validation, Visualization, Writing – review and editing; Hongbin Lu, Conceptualization, Funding acquisition, Supervision, Writing – review and editing

### Author ORCIDs

Hongbin Lu [iD] http://orcid.org/0000-0001-7749-3593

### Ethics

All animal care protocols and experiments in this study were reviewed and approved by the Animal Care and Use Committees of the Laboratory Animal Research Center of our institute. All mice were maintained in the specific pathogen-free facility of the Laboratory Animal Research Center.

### Decision letter and Author response

Decision letter https://doi.org/10.7554/eLife.73614.sa1
Author response https://doi.org/10.7554/eLife.73614.sa2

## Additional files

### Supplementary files

• Transparent reporting form

### Data availability

All data generated or analysed during this study are included in the manuscript and supporting file; Source Data files have been provided for Figures 1 and 2. Figure 3 - Source Data, Figure 4 - Source

Data, Figure 5 - Source Data, Figure 6 - Source, Figure 7 - Source Data, Figure 8 - Source Data contain the numerical data used to generate the figures.

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
