## [Editor Report]

The murine enthesis injury model was used investigate the mechanism of proper mechanical stimulation for enthesis injury repair. Mechanical stimulation could increase the release of active TGF-β1 and enhance mobilization of Prx1+ cells to promote enthesis injury repair via ciliary TGF-β signaling. This work is very significant and will provide an excellent advance in the field.

---

## [Decision Letter]

**Decision letter after peer review:**

Thank you for resubmitting your work entitled "Mechanical stimulation promotes enthesis injury repair by mobilizing Prx1+ cells via ciliary TGF-β signaling" for further consideration by *eLife*. Your article has been reviewed by 2 peer reviewers, including Fayez Safadi as Reviewing Editor and Reviewer #2, and the evaluation has been overseen by Mone Zaidi as Senior Editor.

The manuscript has been improved but there are some issues that need to be addressed, as outlined below:

*Reviewer #1 (Recommendations for the authors):*

The objective of this manuscript is to investigate the role of ciliary TGF-β signaling in the enthesis injury repair, and the mechanical stimulation regulates the release of TGF-β. These findings would be important both with regards to our understanding of ciliary TGF-β signaling and for therapeutical purposes. However, the Prx1+ cells that participated in enthesis development were not the ones that involved in enthesis repair after injury. There is no logic link between Figure 1,2,3a-f and Figure 3g-i. The author should test if active TGF β were also high in the tissues that Prx1+ cells reside in or there is an active TGF β concentration gradient between enthesis and these tissues.

This paper described the role of ciliary TGF-β signaling in promotion of enthesis injury repair and mechanical stimulation enhanced this effect by increase the release of TGF-β. These findings would be important both with regards to our understanding of ciliary TGF-β signaling and for therapeutical purposes. However, there are major issues in the experimental strategies (discussed in more detail below).

1. In Figure 1 and 2, the authors identified Prx1+ cells play a role in enthesis development. They further found Prx1+ cells gave rise to enthesis formation postnatally and were diminished at 12 weeks (Figure 3a-f). However, Prx1+ cell lineage reappeared at the enthesis 4 weeks after RC injury in 12-week-old mice (Figure 3g-i). This data suggested the Prx1+ cells that participated in enthesis development were not the ones that involved in enthesis repair after injury. Therefore, there is no logic link between Figure 1,2,3a-f and Figure 3g-i.

2. Since the authors claimed that there were no Prx1 lineage cells in the adult enthesis, they reasoned that Prx1+ cells that participated in enthesis repair were from other tissues. The authors should identify what tissues that Prx1+ cells reside in in adult mice and are responsible to mechanical loading. It is extremely important because the author later found active TGF β was high after surgery in enthesis where TGF β responding prx1+ cells were rare. The author should test if active TGF β were also high in the tissues that Prx1+ cells reside in or there is an active TGF β concentration gradient between enthesis and these tissues.

3. The authors need confirm whether knockout of cilia from Prx1+ cells will eliminate all TGF-β signaling. Figure 8e clearly showed that knock down PLDN in Prx1+ cells could inhibit TGF-βR2 translocating into the primary cilia, but not all TGF-βR2 on cell membrane.

*Reviewer #2 (Recommendations for the authors):*

The paper entitled "Mechanical stimulation promotes enthesis injury repair by mobilizing Prx1+ cells via ciliary TGF-β signaling authored by Han Xiao et al., described the identification of Prx1+ cells that are involved in enthesis development, those might not be important player in adult enthesis. However, during injury repair, Prx1+ cells could migrate to the injury area to participate in enthesis healing. Proper mechanical stimulation seems to increase the release of TGF-β1 to immobilize Prx1+ cells and promote enthesis injury repair. In addition, ciliary TGF-βR2 was essential for TGF-β signaling transmission during proper mechanical stimulation promoting enthesis injury repair procedure. Data reported in this study are the first to uncover the characteristics of Prx1+ cells on enthesis development and injury healing and provided new insights into the progenitor source for enthesis injury repair. Further, this study found a new mechanism about the mechanical stimulation signal transmission and had pieced together a mechanical conduction mechanism.

The manuscript is well-written and have significant amount of data presented that support the proposed questions. The quality of the data is impressive and convulsive.

Few suggestions were identified,

Figures:

In figure-3 the authors indicated that Prx1+ cells are involved in early development of rotator cuff enthesis. It would be interesting to identify cells that are positive for Prx-1, i.e. are these cell have a stem-like cell phenotype.

Also in figure 3 the authors showed that Prx-1 positive cells are involved in rotator cuff regeneration following injury. It is not the surgical procedure was performed on one side (i.e.) right versus left shoulder. On other word, was the other side served as control?

Figure-4, data presented in figure 4 following injury the stiffness seems to decrease following 30 mins of training, it is not clear why. Also, "e" is missing from the figure legend.

Figure-5, Prx-1positive cells migration was also maximized at 230 mins after training, hence, it is not clear if 20 mins of training will be sufficient for the Prx-1 positive cells to contribute to healing, please discuss this further.

It is very interesting to show that TGF-b mediated healing is regulated by the primary cilia. This conclusion is very interesting and novel findings.

Overall this is a very significant and interesting studies and will further lead to possible proposed therapeutics for rotator cuff injuries and possible other joint injuries.

---

## [Author Response]

Reviewer #1 (Recommendations for the authors):1. In Figure 1 and 2, the authors identified Prx1+ cells play a role in enthesis development. They further found Prx1+ cells gave rise to enthesis formation postnatally and were diminished at 12 weeks (Figure 3a-f). However, Prx1+ cell lineage reappeared at the enthesis 4 weeks after RC injury in 12-week-old mice (Figure 3g-i). This data suggested the Prx1+ cells that participated in enthesis development were not the ones that involved in enthesis repair after injury. Therefore, there is no logic link between Figure 1,2,3a-f and Figure 3g-i.

We are so sorry that our description makes you confused. In this part, we mainly want to reveal the stem cell that was highly participated in the RC injury repair. Until now, which subtype of stem cells essential for RC repair has not been revealed. We thought that revealing the cell atlas during the BTI development would provide us a new sight about the stem cells/progenitors which could take part in BTI repair. So, we committed the single cell sequencing and then found Prx1^+^ cells played an essential role in enthesis development.

To further investigate whether it is involved in injury repair, we used the lineage tracing mice and found that Prx1^+^ cells were disappeared in the 12 weeks old mice, but located at the nearby perichondrium and periosteum. It could reappear after RC injury and participated in the RC repair. The aim of this part study is to look for potential subtype of stem cells/progenitors during the BTI development, which were closely related to the RC repair.

2. Since the authors claimed that there were no Prx1 lineage cells in the adult enthesis, they reasoned that Prx1+ cells that participated in enthesis repair were from other tissues. The authors should identify what tissues that Prx1+ cells reside in in adult mice and are responsible to mechanical loading. It is extremely important because the author later found active TGF β was high after surgery in enthesis where TGF β responding prx1+ cells were rare. The author should test if active TGF β were also high in the tissues that Prx1+ cells reside in or there is an active TGF β concentration gradient between enthesis and these tissues.

Many thanks for your comments and suggestions.

By the lineage tracing study, we found that Prx1^+^ cells were disappeared in the 12 weeks old mice, but located at the nearby perichondrium and periosteum. We have tested the active TGF-β concentration in the perichondrium and found that there is an active TGF β concentration gradient between enthesis and nearby perichondrium. Please view the revised manuscript in line 279-285.

3. The authors need confirm whether knockout of cilia from Prx1+ cells will eliminate all TGF-β signaling. Figure 8e clearly showed that knock down PLDN in Prx1+ cells could inhibit TGF-βR2 translocating into the primary cilia, but not all TGF-βR2 on cell membrane.

Many thanks for your comments and suggestions.

We have knock out the IFT88 in vitro and confirm that knockout of cilia from Prx1+ cells will significantly eliminate TGF-β signaling. Please view the revised manuscript in line 370-375.

Reviewer #2 (Recommendations for the authors):The manuscript is well-written and have significant amount of data presented that support the proposed questions. The quality of the data is impressive and convulsive.Few suggestions were identified,Figures:In figure-3 the authors indicated that Prx1+ cells are involved in early development of rotator cuff enthesis. It would be interesting to identify cells that are positive for Prx-1, i.e. are these cell have a stem-like cell phenotype.

Sincere thanks for your suggestions.

In fact, we have isolated the Prx1^+^ cells and verified that it has the stem-like cell phenotype by flow cytometry and chondrogenesis or osteogenesis induced differentiation experiments. Please view the revised manuscript in line 204-205.

Also in figure 3 the authors showed that Prx-1 positive cells are involved in rotator cuff regeneration following injury. It is not the surgical procedure was performed on one side (i.e.) right versus left shoulder. On other word, was the other side served as control?

Sincere thanks for your comments.

In this part, all the surgical procedure was performed on the left shoulder. When the mouse was sacrificed to harvested the left shoulder, the right shoulder was also harvested to serve as control and verify how the Prx1^+^ cells distribute in the enthesis and the nearby tissue. At the same time, we also harvested the left shoulder in the mouse that the Prx1^+^ cells were labeled by tdTomato and didn’t receive rotator cuff surgery as native group. We found that the Prx1^+^ cells preserve the same distribution characteristic in these two groups without receiving surgery. We have added this information in the revised manuscript in line 574-575.

Figure-4, data presented in figure 4 following injury the stiffness seems to decrease following 30 mins of training, it is not clear why. Also, "e" is missing from the figure legend.

Many thanks for your comments and suggestions.

In this part, we set up four kinds of training parameters and want to establish a reliable mechanical stimulation promoting enthesis repair mouse model to reveal the mechanism behind it. The results showed that the mice in 10 mins and 20 mins groups could benefit from the mechanical stimulation, while the 30 mins group couldn’t. We think mechanical stimulation may have positive or negative effects on enthesis repair. Only proper mechanical stimulation could function as a stimulator for enhancing enthesis repair. The training intensity in the 30 mins group is negative for mouse enthesis repair. Our results showed that both the failure load and stiffness decreased following 30 mins treadmill training. The figure legend of “e” is added. Please view the revised manuscript in line 249-250.

Figure-5, Prx-1positive cells migration was also maximized at 230 mins after training, hence, it is not clear if 20 mins of training will be sufficient for the Prx-1 positive cells to contribute to healing, please discuss this further.

Sincere thanks for your comments.

In figure 5, we found that Prx1^+^ cells migration was maximized at 10 mins and 20 mins groups after training, while few Prx1^+^ cells migrated into the enthesis at 30 mins group. At the same time, we found that mouse in 20 mins group have the best cartilage regeneration and mechanical intensity. Hence, we chosen the 20 mins as the best mechanical training parameter among the 4 groups. We have discussed this further in the revised manuscript. Please view the revised manuscript in line 436-440.

It is very interesting to show that TGF-b mediated healing is regulated by the primary cilia. This conclusion is very interesting and novel findings.Overall this is a very significant and interesting studies and will further lead to possible proposed therapeutics for rotator cuff injuries and possible other joint injuries.

Sincere thanks for your encouragement to our research.